# De novo biosynthesis of the hops bioactive flavonoid xanthohumol in yeast

Shan Yang[1,2], Ruibing Chen [1], Xuan Cao[1], Guodong Wang [3] & Yongjin J. Zhou [1,4,5] ✉

The flavonoid xanthohumol is an important flavor substance in the brewing industry that has a wide variety of bioactivities. However, its unstable structure results in its low content in beer. Microbial biosynthesis is considered a sustainable and economically viable alternative. Here, we harness the yeast *Saccharomyces cerevisiae* for the de novo biosynthesis of xanthohumol from glucose by balancing the three parallel biosynthetic pathways, prenyltransferase engineering, enhancing precursor supply, constructing enzyme fusion, and peroxisomal engineering. These strategies improve the production of the key xanthohumol precursor demethylxanthohumol (DMX) by 83-fold and achieve the de novo biosynthesis of xanthohumol in yeast. We also reveal that prenylation is the key limiting step in DMX biosynthesis and develop tailored metabolic regulation strategies to enhance the DMAPP availability and prenylation efficiency. Our work provides feasible approaches for systematically engineering yeast cell factories for the de novo biosynthesis of complex natural products.

Hops (*Humulus lupulus* L., Cannabaceae), an essential ingredient in beer brewing, contain many secondary metabolites, including essential oils, bitter acids and prenylated flavonoids[1,2]. Among these metabolites, xanthohumol is a prenylated flavonoid that has a variety of biological activities, such as the prevention of cancer, the prevention and mitigation of diabetes, and antioxidant, anti-inflammatory, antibacterial and immunomodulatory effects[3-7]. However, the low concentration (0.1–1% dry cell weight) and instability of xanthohumol during beer brewing do not allow the pharmacological and health effects of xanthohumol to be obtained through beer ingestion[8]. Furthermore, xanthohumol extraction from hops faces material shortages due to geographical limitations, and the leftover solid waste[9] and solvent use[10,11] will bring environmental stress.

Alternatively, microbial biosynthesis provides a versatile approach for the sustainable production of scarce natural products due to the great developments in metabolic engineering and synthetic

biology[12,13]. The brewing yeast *Saccharomyces cerevisiae* is considered an ideal host for the production of phyto-natural products, including terpenoids, alkaloids, and flavonoids[14-16]. Thus, engineering *S. cerevisiae* for such biosyntheses might provide a sustainable route to supply large amounts of xanthohumol and enhance the xanthohumol level in beer, since engineering biosynthesis of monoterpene linalool and geraniol in brewing yeast gives rise to a hoppy beer flavor[17]. In particular, engineering the production of xanthohumol in brewing yeast not only give the full flavor of beer, and also provide sufficient supply of xanthohumol for functional food and pharmaceuticals. However, the complexity of biosynthetic pathways and the tight regulation of precursor supply make it challenging to reconstruct the de novo biosynthesis of xanthohumol in yeast cell factories. Indeed, it have not yet been any reports on microbial xanthohumol biosynthesis.

In recent decades, considerable efforts have been made to identify the key enzymes in the complete biosynthetic pathway of

[1]Division of Biotechnology, Dalian Institute of Chemical Physics, Chinese Academy of Sciences, Dalian 116023, China. [2]University of Chinese Academy of Sciences, Beijing 100049, China. [3]State Key Laboratory of Plant Genomics, Institute of Genetics and Developmental Biology, The Innovative Academy of Seed Design, Chinese Academy of Sciences, Beijing 100101, China. [4]CAS Key Laboratory of Separation Science for Analytical Chemistry, Dalian Institute of Chemical Physics, Chinese Academy of Sciences, Dalian 116023, China. [5]Dalian Key Laboratory of Energy Biotechnology, Dalian Institute of Chemical Physics, Chinese Academy of Sciences, Dalian 116023, China. ✉e-mail: zhouyongjin@dicp.ac.cn

xanthohumol[2,18–21]. To facilitate the construction and optimization of xanthohumol biosynthesis in yeast, we modularly mapped the biosynthetic pathway from glucose, which comprises four modules: *p*-coumaric-CoA (*p*-CA-CoA) biosynthesis (Module I), malonyl-CoA supply (Module II), prenylation (Module III) and methylation (Module IV) (Fig. 1). The parallel biosynthesis of the three precursors, *p*-CA-CoA, malonyl-CoA and dimethylallyl pyrophosphate (DMAPP), requires careful balancing and tuning of the biosynthetic pathways and is distinct from common upstream to downstream pathways, such as terpenoid biosynthesis (Supplementary Fig. 1)[22].

In this work, we engineer a yeast platform for the de novo biosynthesis of xanthohumol by balancing the three parallel aromatic biosynthesis, malonyl-CoA supply, and the mevalonate (MVA) pathway. We then enhance the limiting prenylation step by selecting and engineering prenyltransferase (PTase) and enhancing the DMAPP availability, which enables an 83-fold improvement in the demethylxanthohumol (DMX) production (4.0 mg/L). Finally, optimization of the last methylation step achieves the de novo biosynthesis of xanthohumol from glucose in yeast. This microbial biosynthesis of xanthohumol will allow microbial cell factories to be optimized for the sustainable supply of xanthohumol and provide feasible approaches to optimize other complex biosynthetic pathways of natural products.

## Results

### Modular construction of the xanthohumol biosynthetic pathway

First, we modularly constructed the biosynthetic pathway to produce DMX, the key precursor of xanthohumol. We first constructed the downstream pathway of DMX biosynthesis starting with L-tyrosine by expressing the tyrosine ammonia-lyase gene *FjTAL* from *Flavobacterium johnsoniae*[23], the 4-coumarate-coenzyme A ligase gene *HlCCL1*[2], the chalcone synthase gene *CHS_H1*[21] and the prenyltransferase gene *HlPT1L*[19] from *H. lupulus* (Fig. 2a). We also expressed a noncatalytic chalcone isomerase gene, *HlCHIL2*, from *H. lupulus*, which has been shown to enhance the activities of CHS_H1 and HlPT1L[18]. In addition, the *H. lupulus O*-methyltransferase gene *HlOMT1*[20] was expressed to examine the possible xanthohumol production from DMX. The resulting strain YS103 produced 25 mg/L naringenin chalcone (NC) and naringenin (N, a product spontaneously formed product from the unstable compound NC[24]), but neither xanthohumol nor DMX (Fig. 2b, d).

We then tried to optimize the upstream biosynthetic pathway to enhance the synthesis of the precursors L-tyrosine, malonyl-CoA and DMAPP. We first tried to enhance L-tyrosine biosynthesis by knocking out gene *ARO10*, which encodes the competing enzyme phenylpyruvate decarboxylase, as well as overexpressing genes *ARO4*[K229L] (encoding a feedback-inhibition resistant version of DAHP synthase)

**Fig. 1 | Engineered metabolic pathway for the de novo biosynthesis of xanthohumol in yeast.** The DMX biosynthetic pathway was derived from three endogenous module pathways, including the aromatic pathway (Module I, purple label), malonyl-CoA pathway (Module II, orange label), and MVA pathway (Module III, blue label). Then, DMX was methylated to form xanthohumol (Module IV, pink label). Exogenous enzymes are also labeled in pink and competing metabolic pathways are labeled in gray. Dotted arrows represent multiple steps. E4P erythrose 4-phosphate, PEP phosphoenolpyruvate, DAHP 3-deoxy-D-arabino-heptulosonic acid 7-phosphate, EPSP 5-enolpyruvyl-shikimate-3-phosphate, CHA chorismic acid, PPA prephenate, PPY phenylpyruvate, HPP para-hydroxy-phenylpyruvate, *p*-PAC para-hydroxy-acetaldehyde, L-TYR L-tyrosine, FFA free fatty acid, PYR pyruvate, HMG-CoA 3-hydroxy-3-methylglutaryl coenzyme A, MVA mevalonate, MVA-P mevalonate-5-phosphate, MVA-PP mevalonate-5-pyrophosphate, DMAPP dimethylallyl diphosphate, IPP

isopentenyl pyrophosphate, FPP farnesyl pyrophosphate, SAM S-adenosyl-L-methionine, *ARO1* encoding shikimate dehydrogenase, *ARO2* encoding chorismate synthase, *ARO4* encoding DAHP synthase, *ARO7* encoding chorismate mutase, *ARO8* encoding aromatic aminotransferase I, *ARO9* encoding aromatic aminotransferase II, *ARO10* encoding phenylpyruvate decarboxylase, *ACC1* encoding acetyl-CoA carboxylase, *ERG10* encoding acetoacetyl-CoA thiolase, *ERG13* encoding HMG-CoA synthase, *HMGR* encoding HMG-CoA reductase, *ERG12* encoding mevalonate kinase, *ERG8* encoding phosphomevalonate kinase, *ERG19* encoding mevalonate decarboxylase, *IDI1* encoding isopentenyl diphosphate isomerase, *ERG20* encoding farnesyl diphosphate synthase, *FjTAL* encoding tyrosine ammonia-lyase, *HlCCL1* encoding 4-coumarate-coenzyme A ligase, *CHS_H1* encoding chalcone synthase, *HlCHIL2* encoding noncatalytic chalcone isomerase, *HlPT1L* encoding prenyltransferase, *HlOMT1* encoding *O*-methyltransferase.

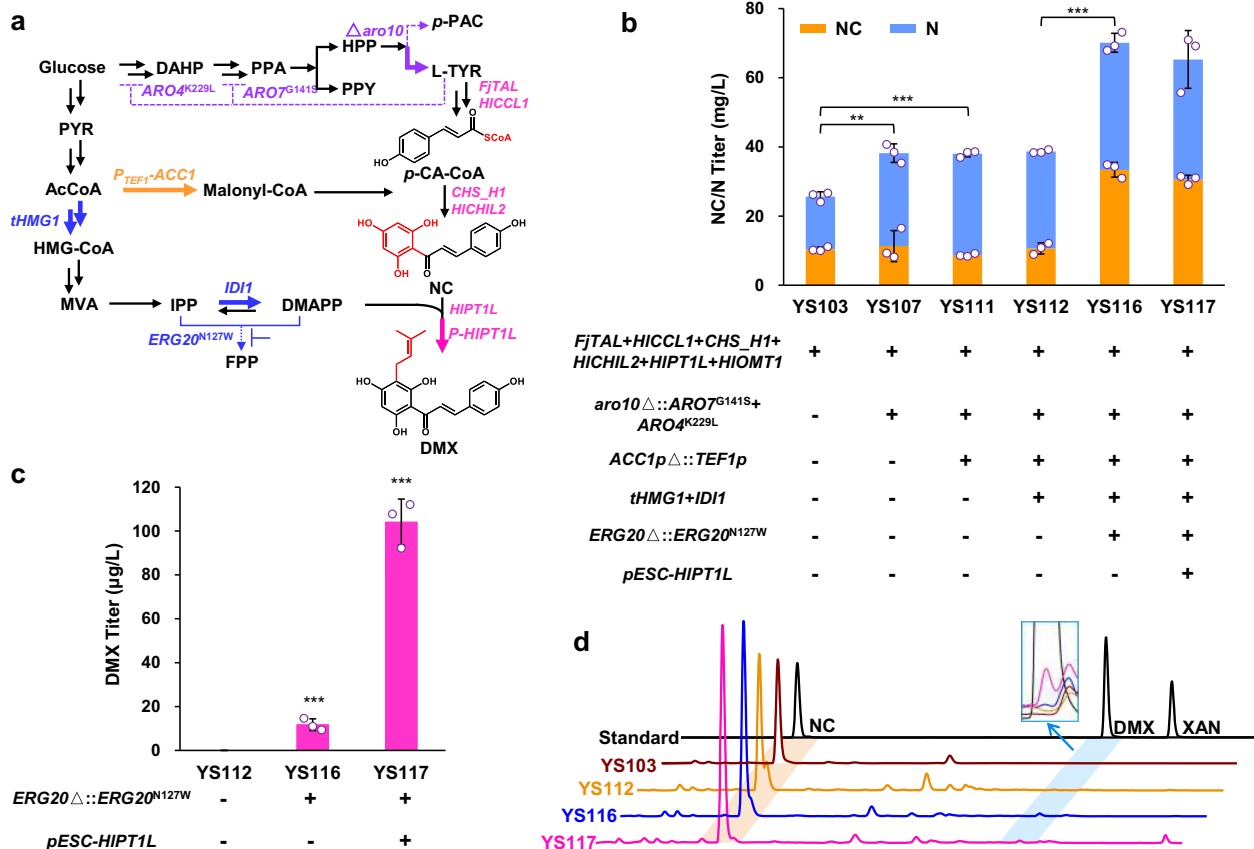

**Fig. 2 | Optimizing the DMX de novo biosynthetic pathway. a** Schematic overview of the modification of three parallel biosynthetic modules. The three modules include the aromatic pathway (purple label), malonyl-CoA pathway (orange label), and MVA pathway (blue label). Upregulated steps are indicated with bold arrows, and downregulated or knockout steps are shown with dashed arrows. *FjTAL* encoding tyrosine ammonia-lyase, *HlCCL1* encoding 4-coumarate-coenzyme A ligase, *CHS_H1* encoding chalcone synthase, *HlCHIL2* encoding noncatalytic chalcone isomerase, *HlPT1L* encoding prenyltransferase, *aro10* encoding phenylpyruvate decarboxylase, *ARO4^{K229L}* and *ARO7^{G141S}* encoding resistant versions of DAHP synthase and chorismate mutase, *ACC1* encoding acetyl-CoA carboxylase, *tHMG1* encoding truncated HMG-CoA reductase 1, *IDI1* encoding isopentenyl diphosphate

isomerase, *ERG20^{N127W}* encoding variant of farnesyl diphosphate synthase, P-HlPT1L, overexpression of prenyltransferase by the pESC-*URA* plasmid. See Fig. 1 legend regarding other abbreviations. **b** Metabolic modification of these three modules improved the production of precursors NC/N. **c** Mutation of ERG20 and overexpression of HlPT1L improved DMX production. **d** HPLC analysis of the DMX standard and the fermented product of strains YS103, YS112, YS116 and YS117. All strains were cultivated in 100 mL shake flasks containing 20 mL of minimal medium. Mean values ± standard deviations are shown (*n* = 3 independent biological samples). Student's *t*-test was used for comparing two groups (*$p < 0.05$, **$p < 0.01$, ***$p < 0.001$), and *p* values were shown in **b**, **c**. Source data are provided as a Source Data file.

and *ARO7^{G141S}* (encoding a chorismate mutase)[25]. The resulting strain YS107 produced 52% more NC/N than that of the parent strain YS103 (Fig. 2b). We then tried to improve malonyl-CoA biosynthesis by replacing the native promoter of the acetyl-CoA carboxylase gene (*ACC1*) with the constitutive promoter $P_{TEF1}$[26]. This replacement failed to improve the production of NC/N, suggesting that the acetyl-CoA carboxylation was not currently a bottleneck.

Finally, we attempted to enhance prenylation step by enhancing the supply of the low-level substrate DMAPP and overexpressing of the PTase gene. Overexpressing the MVA rate-limiting genes *tHMG1* (encoding truncated HMG-CoA reductase 1) and *IDI1* (encoding isopentenyl diphosphate isomerase) still failed to produce DMX in strain YS112. We speculated that the efficient process of farnesyl diphosphate (FPP) biosynthesis competed with the biosynthesis with DMAPP, and it has been reported that the FPP synthase mutation Erg20^{N127W} resulted in decreased activity in catalyzing DMAPP turnover[27]. Thus, we mutated *ERG20* in situ to *ERG20^{N127W}* to improve the DMAPP availability, which succeeded in producing 12 μg/L DMX in strain YS116 (Fig. 2c, d). This data suggested that blocking DMAPP consumption was more essential for DMX biosynthesis than enhancing DMAPP supply. Interestingly, this modification improved NC/N production by 79% (Fig. 2b), which might be due to decrease the MVA flux toward FPP biosynthesis,

and thus enhanced the acetyl-CoA availability for malonyl-CoA supply. However, strain YS116 had decreased maximum specific growth rate (μ_{max}) compared to parent strain YS112 (Supplementary Fig. 2). These data suggested that mutation of FPPS increased the availability of prenyl moiety DMAPP and further retarded cell growth. Subsequently, we attempted to overexpress HlPT1L by using a high-copy plasmid, and the resulting strain YS117 produced 7.7-fold more DMX (104 μg/L) than the parent strain YS116 (Fig. 2c, d), which indicated that PTase was a limiting factor in DMX production.

## Characterizing and engineering PTase
We further tried to enhance PTase activity to improve DMX production (Fig. 3a). It has been reported that membrane PTase is targeted to plastids with a high pH in plants[19], while the yeast cytosolic pH is low (<6) due to the accumulation of acidic metabolites[28]. Therefore, we cultivated the engineered *S. cerevisiae* YS116 with MES buffered media, which maintained the pH of the media at 5.4 for 72 h of cultivation, while the pH of the control media was 3.3. The DMX production in the buffered medium was 2.8-fold higher than that in the nonbuffered medium (Fig. 3b). However, there was complete transformation of the precursor NC into N, indicating that the high pH in the cytosol made it difficult to stabilize the ring-opened conformation of NC (Fig. 3b).

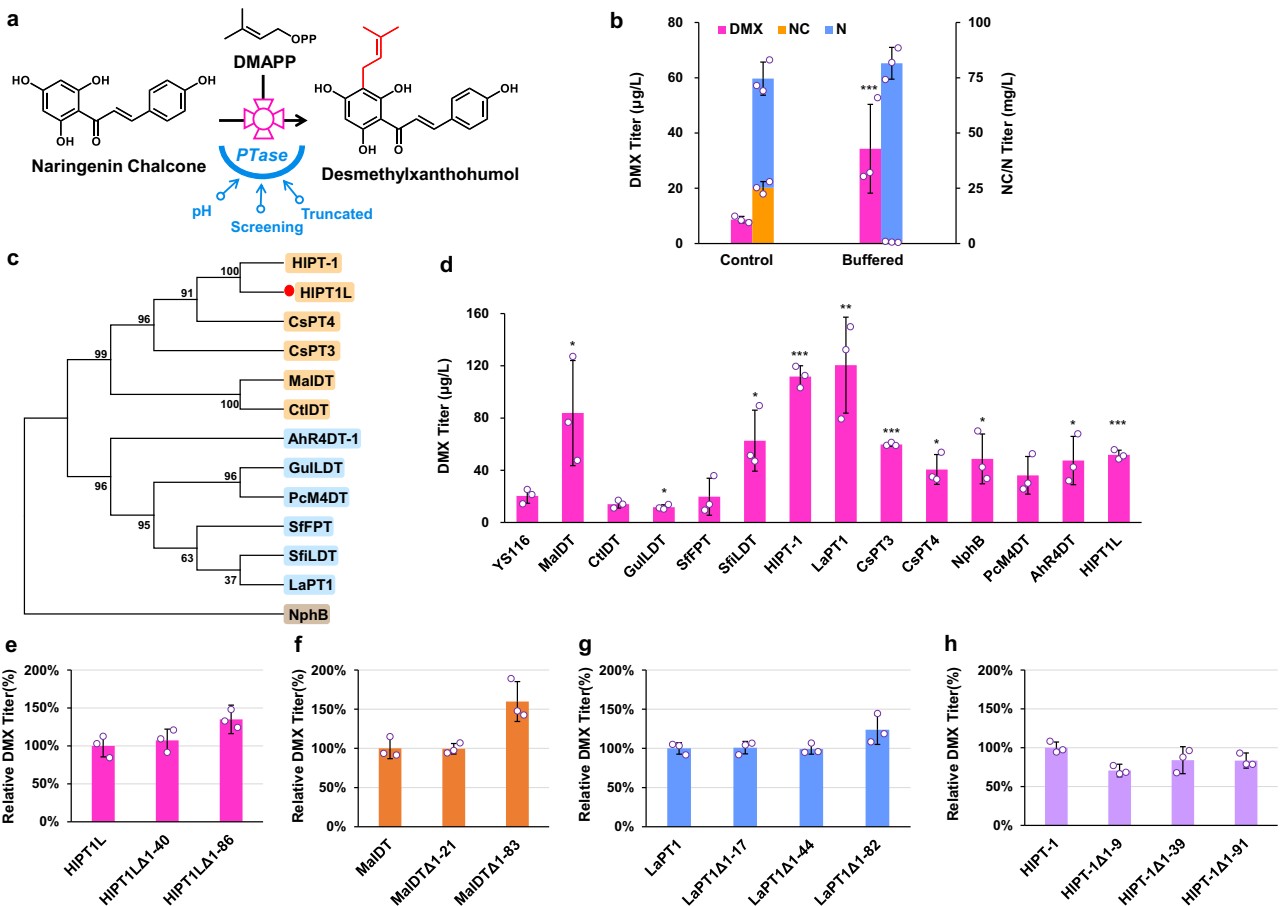

**Fig. 3 | Engineering the PTase to enhance DMX production. a** Schematic illustration of PTase engineering. To improve the catalytic ability of PTase by changing the pH, screening the alternative PTase and truncating the signal peptides of PTase. **b** Adding MES buffer solution increased DMX production in strain YS116. MES buffer with a pH value of 6.58 was added after 24 h of fermentation. **c** Phylogenetic analysis of PTases. A neighbor-joining tree was constructed by using MEGA7 software and a maximum likelihood method with 1000 bootstrap tests.

**d** Evaluating different PTases in strain YS116 by using high-copy plasmids. **e–h** Truncating PTases for DMX production in strain YS116. All strains were cultivated in 100 mL shake flasks containing 20 mL of minimal medium. Mean values ± standard deviations are shown ($n = 3$ independent biological samples). Student's $t$-test was used for comparing two groups (*$p < 0.05$, **$p < 0.01$, ***$p < 0.001$), and $p$ values were shown in **b**, **d**. Source data are provided as a Source Data file.

We then searched for alternative PTases from hops and other organisms based on similarities of the substrate flavonoids and prenyl donors, including hops HlPT-1[29] with 98.5% homology to HlPT1L, four chalcone-specific PTases that catalyze the condensation of isoliquiritigenin and DMAPP (MaIDT[30] and CtIDT[30] from the Moraceae family and GuILDT[31] and SfiLDT[32] from the Leguminosae family), four isoflavone-specific PTases (SfFPT[33], LaPT1[34], AhR4DT-1[35] and PcM4DT[36]), *Cannabis sativa L.* CsPT3[37] and CsPT4[38], and a soluble PTase NphB[39] from *Streptomyces sp.* An overview of all examined PTases and their substrate specificities are presented in Supplementary Data 1. Amino acid sequence alignment of these 13 PTases was performed to construct a phylogenetic tree and to evaluate the evolutionary relationship between HlPT1L and these other PTases (Fig. 3c). The results showed that HlPT1L and CsPT4 were in the same branch, indicating that their enzymatic catalytic functions were similar. HlPT1L was also closely evolutionarily related to MaIDT, CtIDT and CsPT3 but was far removed from other PTase enzymes.

We expressed these 12 codon-optimized PTases and HlPT1L in strain YS116 by using a high-copy plasmid (Fig. 3d). When MaIDT, HlPT-1 and LaPT1 were expressed, more DMX was produced than when HlPT1L was expressed, among which LaPT1 led to the highest DMX production of 121 µg/L (Fig. 3d). We then truncated the N-terminal signal peptide to improve the enzyme activity with the aid of the signal

peptide prediction software TargetP (https://services.healthtech.dtu.dk/services/TargetP-2.0/), and the truncated sequence positions are shown in Supplementary Fig. 3. Subsequently, the truncated PTase sequences, together with the full-length sequences, were transferred into strain YS116 to evaluate DMX production. Truncation of HlPT1L, MaIDT and LaPT1 improved DMX production (Fig. 3e–g), while truncation of HlPT-1 led to lower DMX production than the original version (Fig. 3h). Finally, truncated HlPT1L$_{\Delta 1-86}$ was used for further experiments.

## Pathway optimization to improve DMX biosynthesis

Next, we tried to optimize the biosynthetic pathways to improve precursor supply (Fig. 4a). We first optimized the MVA pathway to supply the precursor DMAPP. *ERG20*$^{N127W}$ was downregulated to further decrease the turnover of DMAPP, by replacing its native promoter with the *HXT1* promoter[40] (strain YS119) or *ERG1* promoter[41] (strain YS120), and DMX production was improved by 2-fold and 1.4-fold than compared with the parent strain YS116, respectively (Fig. 4b). P$_{HXT1}$ is a high glucose-induced and low glucose-repressed promoter that should be helpful when synchronizing DMX biosynthesis with the modified GAL regulation system[42]. However, these two replacements not only failed to increase the NC/N production but also reduced the cell biomass (Supplementary Fig. 4a). We also overexpressed genes *ERG10* encoding acetoacetyl-CoA thiolase and *HMG2*$^{K6R}$ encoding a

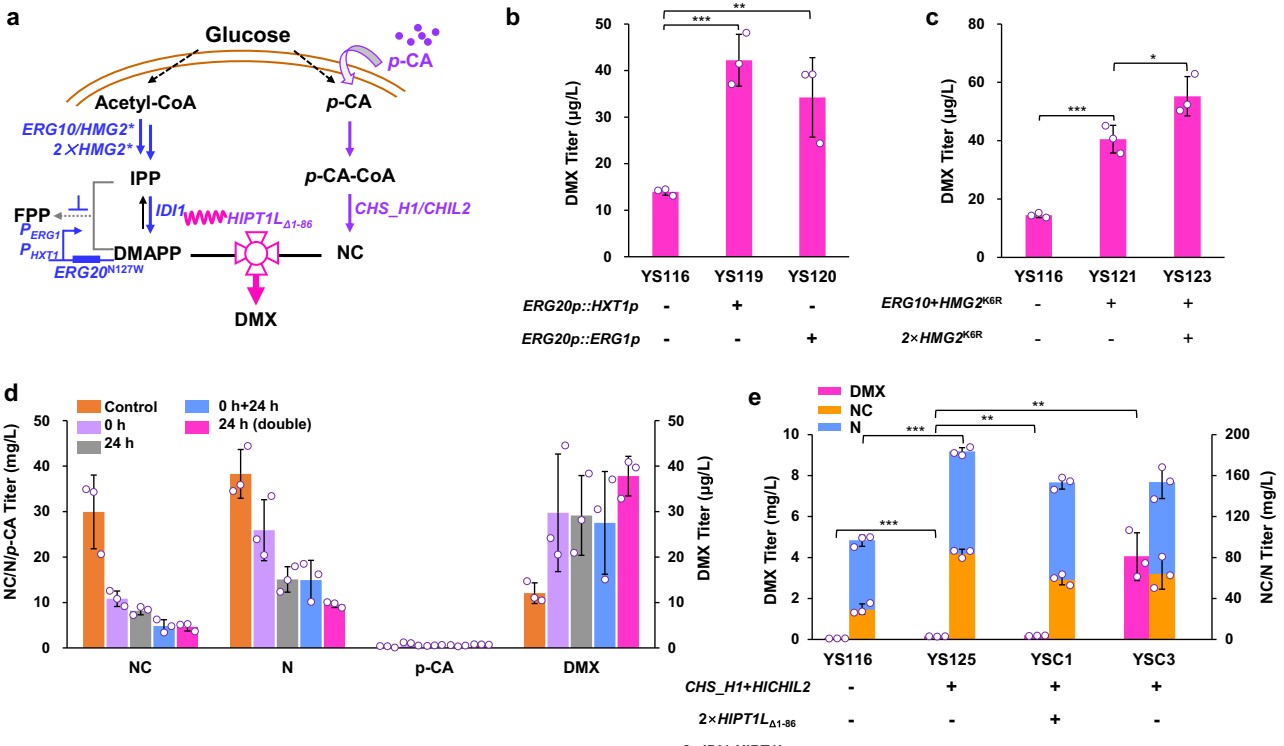

**Fig. 4 | Engineering substrate supply and enzyme fusion to enhance DMX production. a** Schematic diagram of substrate supply and enzyme fusion. Enzyme overexpression is shown in purple and blue, and downregulated expression is indicated in gray. The supply of DMAPP was increased by overexpressing *ERG10* and *HMG2^{K6R}* and replacing the native promoter of *ERG20^{N127W}* with the promoter $P_{HXT1}$ or $P_{ERG1}$. The supply of NC was enhanced by feeding *p*-CA or overexpressing *CHS_H1* and *HlCHIL2*. IDI1-HlPT1L$_{Δ1-86}$ fusion protein with the linker (GGGS)$_3$ enhanced DMX production. **b** Replacing the native promoter of *ERG20^{N127W}* with the promoter $P_{HXT1}$ or $P_{ERG1}$ improved DMX production. **c** Overexpression of *ERG10* and

*HMG2^{K6R}* increased DMX production. **d** Addition of *p*-CA at different times to produce DMX. For 0 h and 24 h, 140 mg/L *p*-CA was added at 0 h and 24 h, respectively. For 0 h + 24 h, 140 mg/L *p*-CA was added at both 0 h and 24 h. For 24 h (double), 280 mg/L *p*-CA was added at 24 h. **e** Overexpression of *CHS_H1*, *HlCHIL2* or *IDI1-HlPT1L$_{Δ1-86}$* fusion increased DMX production. All strains were cultivated in 100 mL shake flasks containing 20 mL of minimal medium. Mean values ± standard deviations are shown ($n = 3$ independent biological samples). Student's *t*-test was used for comparing two groups (*$p < 0.05$, **$p < 0.01$, ***$p < 0.001$), and *p* values were shown in **b**, **c**, **e**. Source data are provided as a Source Data file.

mutant of endogenous HMG-CoA reductase (strain YS121)[43], which improved DMX production by 1.9-fold compared to control strain YS116. The genome integrating another two copies of *HMG2^{K6R}* (strain YS123) further improved DMX production by 34% compared with strain YS121. These results indicated that overexpressing rate-limiting genes in the MVA pathway could increase the supply of DMAPP, thereby increasing DMX production. However, we found that the amount of precursor NC/N in strain YS123 was slightly reduced compared with that in strain YS121 (Supplementary Fig. 4b), indicating that continued overexpression of MVA pathway genes would lead to excessive metabolic flux into the MVA pathway from acetyl-CoA, thus reducing malonyl-CoA biosynthesis. This observation was consistent with that down-regulation of FPP biosynthesis improved NC/N production (strain YS116 in Fig. 2b) by possible increasing the flux of acetyl-CoA to malonyl-CoA. Therefore, we tried to construct an isopentenol utilization pathway (IUP)[44] to supply DMAPP by alleviating the metabolic stress on central carbon metabolism. However, expression of the IUP with isopentenol supplementation failed to improve DMX production and decreased the biosynthesis of precursor NC/N (Supplementary Fig. 5).

Since *p*-coumaric acid (*p*-CA) was not detected in the strain, we speculated that increasing the *p*-CA supply could further improve DMX production. Feeding different concentrations of *p*-CA was helpful to increase DMX production (Supplementary Fig. 6), among which feeding 280 mg/L *p*-CA at 24 h improved DMX production by 2.2-fold (Fig. 4d). Interestingly, there was almost no *p*-CA remaining, suggesting that all of the *p*-CA was converted into *p*-CA-CoA. Therefore, we

tried to enhance the turnover of *p*-CA-CoA toward NC biosynthesis by overexpressing *CHS_H1* and *HlCHIL2* in strain YS116. The resulting strain YS125 produced 2.1-fold and 1.8-fold more DMX and NC than the parent strain YS116, respectively (Fig. 4e), suggesting that overexpressing *CHS_H1* and *HlCHIL2* drives metabolic flux toward NC biosynthesis and DMX production.

Although an increase in the supply of DMAPP and NC improved DMX production, the DMX titer was still low (<150 μg/L). Considering that the presence of a signal peptide prevented the PTase enzyme from quickly contacting the substrates DMAPP and NC, we further expressed two extra copies of *HlPT1L$_{Δ1-86}$*, which improved DMX production to 184 μg/L in strain YSC1 (Fig. 4e). Unfortunately, the introduction of *HlPT1L$_{Δ1-86}$* increased DMX production by only 23% compared to the strain YS125. We found that the total production of NC/N in strain YSC1 reached 153 mg/L, indicating that there was sufficient amount of the precursor NC in the cytosol. Since IDI1 catalyzed the isomerization of IPP into DMAPP, we fused IDI1 with HlPT1L$_{Δ1-86}$ to help the quick interaction of DMAPP with PTase. Genomic integration of two copies of the fusion gene *IDI1-HlPT1L$_{Δ1-86}$* resulted in DMX production of 4 mg/L in strain YSC3, which was 83-fold and 21-fold higher than that in strains YS116 and YSC1, respectively (Fig. 4e). These results showed that the IDI1-HlPT1L$_{Δ1-86}$ fusion protein strategy could indeed allow quick contact between DMAPP and the PTase enzyme and improve the production of DMX. Due to the significant effect of this fusion strategy, we further transferred two extra copies of the fusion gene *IDI1-HlPT1L$_{Δ1-86}$* into strain YSC3 to obtain strain YSC4 (Supplementary Fig. 7), hoping to further increase DMX production. Moreover, because

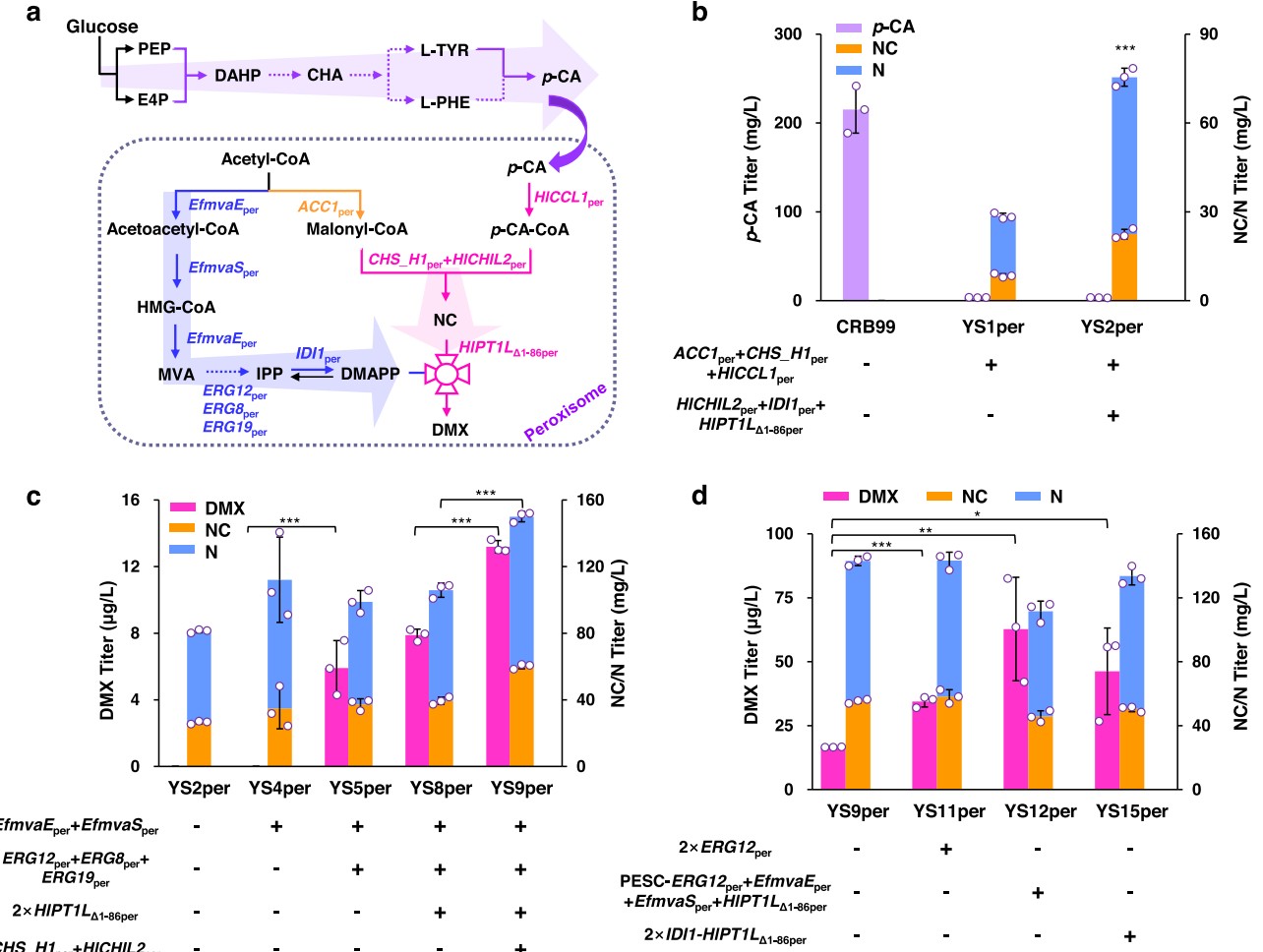

**Fig. 5 | Engineering DMX biosynthesis in peroxisomes. a** Overview of the engineered metabolic pathway for DMX biosynthesis in peroxisomes. Precursor *p*-CA was efficiently produced in the cytosol. The 3-step reaction from *p*-CA to DMX was targeted to the peroxisome in the *p*-CA overproducing chassis RB99 (pink label). The malonyl-CoA and MVA pathway modules (orange and blue label) were also targeted to the peroxisomes. *EfmvaE*$_{per}$ encoding acetoacetyl-CoA thiolase/HMG-CoA reductase, *EfmvaS*$_{per}$ encoding HMG-CoA synthase, *ERG12*$_{per}$ encoding mevalonate kinase, *ERG8*$_{per}$ encoding phosphomevalonate kinase, *ERG19*$_{per}$ encoding mevalonate diphosphate decarboxylase, *IDI1*$_{per}$ encoding isopentenyl diphosphate isomerase, *ACC1*$_{per}$ encoding acetyl-CoA carboxylase, *HlCCL1*$_{per}$ encoding 4-coumarate-coenzyme A ligase, *CHS_H1*$_{per}$ encoding chalcone synthase, *HlCHIL2*$_{per}$

encoding noncatalytic chalcone isomerase, *HlPT1L*$_{\Delta1-86per}$ encoding truncated prenyltransferase. **b** Engineering peroxisomal malonyl-CoA and DMX biosynthetic modules for synthesis of the precursor NC/N. **c** Construction of the complete MVA pathway in the peroxisome enabled de novo synthesis of DMX and increased DMX production by enhancing PTase expression and NC supply. **d** Optimizing the MVA pathway and overexpressing *IDI1-HlPT1L*$_{\Delta1-86per}$ increased DMX production. All strains were cultivated in 100 mL shake flasks containing 20 mL of minimal medium. Mean values ± standard deviations are shown (*n* = 3 independent biological samples). Student's *t*-test was used for comparing two groups (**p* < 0.05, ***p* < 0.01, ****p* < 0.001). Source data are provided as a Source Data file.

HlCHIL2 could enhance the activity of the HlPT1L enzyme, we tried to fuse HlCHIL2 with HlPT1L$_{\Delta1-86}$ to improve DMX production. Unfortunately, we found that the addition of these two fusion proteins decreased DMX production by 77% (IDI1-HlPT1L$_{\Delta1-86}$) and 65% (HlCHIL2-HlPT1L$_{\Delta1-86}$), respectively, compared with strain YSC3 (Supplementary Fig. 7).

## Peroxisome compartmentalization for DMX biosynthesis

We showed that the availability of substrate DMAPP was critical for the prenylation reaction during DMX biosynthesis. Sub-organelle compartmentalization is helpful for selective production, as it relieves the competition with cytosolic enzymes[45]. Peroxisomes are ideal compartments with an efficient supply of acetyl-CoA from fatty acid β-oxidation and the absence of fatty acid biosynthesis and FPP competition[46–49], which might be helpful to accumulate the DMX precursors DMAPP and malonyl-CoA. We thus compartmentalized the DMX downstream pathway from *p*-CA into peroxisomes with reconstruction of the peroxisomal DMAPP biosynthetic

pathways (Fig. 5a). We used a previously constructed *p*-CA-overproducing (131 mg/L) strain RB14 as a chassis[42] and then overexpressed *ARO1* (encoding shikimate dehydrogenase), *ARO2* (encoding chorismate synthase) and *ARO3* (encoding DAHP synthase) in strain RB14 to obtain strain RB99 to further improve *p*-CA production. Peroxisomal targeting of endogenous *ACC1* (*ACC1*$_{per}$) and exogenous *CHS_H1*$_{per}$ and *HlCCL1*$_{per}$ (strain YS1per) resulted in NC/N production of 28 mg/L (Fig. 5b). Further peroxisomal construction of prenylation through the expression of *IDI1*$_{per}$, *HlPT1L*$_{\Delta1-86per}$ and *HlCHIL2*$_{per}$ failed to produce DMX in the engineered strain YS2per (Fig. 5b, c).

To provide a sufficient pool of the precursor DMAPP for DMX production, we reconstructed the MVA pathway in peroxisomes. We expressed *Enterococcus faecalis* EfmvaE$_{per}$ (bifunctional acetoacetyl-CoA thiolase/HMG-CoA reductase) and EfmvaS$_{per}$ (HMG-CoA synthase)[50] to catalyze the first three steps of the MVA pathway due to their high efficiency and absence of feedback regulation in yeast. *ERG12*$_{per}$, *ERG8*$_{per}$ and *ERG19*$_{per}$ were also expressed in peroxisomes;

## a

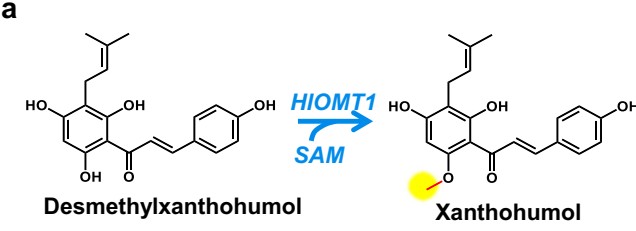

**Desmethylxanthohumol**          **Xanthohumol**

## b

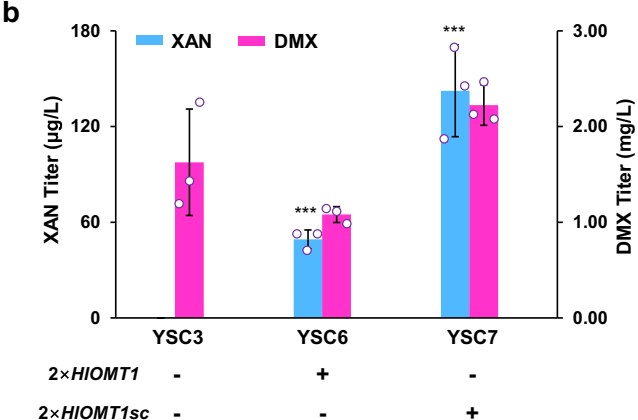

**Fig. 6 | De novo biosynthesis of xanthohumol. a** Scheme of xanthohumol production from DMX by an O-methyltransferase (HlOMT1). **b** Overexpressing *HlOMT*1 and *HlOMT*1sc (codon-optimized for *S. cerevisiae*) for xanthohumol (XAN) production. All strains were cultivated in 100 mL shake flasks containing 20 mL of minimal medium. Mean values ± standard deviations are shown (*n* = 3 independent biological samples). Student's *t*-test was used for comparing two groups (**p* < 0.05, ***p* < 0.01, ****p* < 0.001). Source data are provided as a Source Data file.

however, the engineered strain YS5per with the peroxisomal MVA pathway led to the production of a low DMX titer (6 µg/L). We then enhanced the prenylation step by adding another two copies of peroxisomal *HlPT1L*$_{Δ1-86per}$ (strain YS8per), which slightly improved DMX production (Fig. 5c). Subsequently, *CHS_H1*$_{per}$ and *HlCHIL2*$_{per}$ were expressed in strain YS8per, and the resulting strain YS9per produced 63% more DMX than the parent strain YS8per (Fig. 5c).

After enhancement of the prenylation step, we further tried to increase the DMAPP supply by expressing another two copies of the most essential gene *ERG12*$_{per}$[47] (strain YS11per), which improved DMX production by 100% (Fig. 5d). This improvement encouraged us to further optimize the peroxisomal MVA pathway. Thus, *ERG12*$_{per}$, *EfMVAE*$_{per}$ and *EfMVAS*$_{per}$ were overexpressed in a high-copy plasmid co-expressing *HlPT1L*$_{Δ1-86per}$, and the resulting strain YS12per produced 2.7-fold more DMX than strain YS9per (Fig. 5d). We also tried to express the fusion gene *IDI1-HlPT1L*$_{Δ1-86per}$ to channel DMAPP toward PTase (strain YS15per), which improved DMX production by 1.7-fold compared to strain YS9per. However, the DMX titers were still low (<100 µg/L) and were much lower than those of the cytosolic pathway (4.0 mg/L in strain YSC3). This low DMX production might be attributed to the low activity of the PTase enzyme and/or low DMAPP availability in peroxisomes.

### Biosynthesis of xanthohumol

We finally tried to biosynthesize xanthohumol from DMX by expressing the O-methyltransferase gene *HlOMT1* from *H. lupulus* in the chassis YSC3 harboring the cytosolic DMX biosynthetic pathway (Fig. 6a). We expressed the original version of *HlOMT1* and codon-optimized version *HlOMT1*sc for expression in *S. cerevisiae* in the chassis strain YSC3. The resulting strains YSC6 (*HlOMT1*) and YSC7 (*HlOMT1*sc) produced 49 µg/L and 142 µg/L xanthohumol (Fig. 6b),

respectively, and liquid chromatography-mass spectrometry analysis verified xanthohumol production (Supplementary Fig. 8). The high accumulation of DMX (1.1–2.2 mg/L) suggests that the methylation step should be further enhanced for efficient xanthohumol production from DMX.

## Discussion

Hops (*H. lupulus* L.) are valuable sources of several secondary metabolites, such as essential oils, bitter acids and flavonoids, which have potential medical applications. Microbial synthesis is considered a feasible approach for the efficient production of low-content natural products such as bitter acids and essential oils from hops[17,51]. Xanthohumol is a functional flavonoid in beer that has a variety of pharmacological effects. However, its complex biosynthesis poses challenges for its de novo production in microbes. In this study, we systematically engineered the budding yeast *S. cerevisiae* for de novo xanthohumol biosynthesis by optimizing the biosynthetic pathway and rewiring the cellular metabolism.

To facilitate pathway construction and optimization, we divided the reconstructed metabolism into three modules: the *p*-CA-CoA, malonyl-CoA and MVA biosynthetic pathways. Here, we identified that the prenylation of NC with DMAPP was a limiting step in DMX biosynthesis, since the DMX level was much lower than the precursor NC in engineered strain. Previous studies also showed that the prenylation was a limiting step in biosynthesis of prenylated compounds[38,52,53]. Thus, the prenylation step should be enhanced to improve DMX production. Here, we applied enzyme discovery, truncation of the signal peptide and enhancement of expression levels to improve PTase activity, which significantly improved DMX biosynthesis from NC.

In addition to PTase activity, the limited availability of DMAPP is another bottleneck for efficient prenylation[54,55], since DMAPP level is tightly regulated and efficiently transformed toward FPP in ergosterol biosynthesis in yeast[56]. We here developed several pathway engineering strategies to enhance the DMAPP supply and enzyme fusion improve the DMAPP availability to PTase. To improve DMAPP supply, we reduced DMAPP consumption toward FPP by expressing an FPPS mutant gene *ERG20*$^{N127W}$, downregulating the expression of *ERG20*$^{N127W}$ and overexpressing key rate limiting MVA genes for enhanced upstream flux. These strategies improved the DMX production. However, it was still lower than 0.1 mg/L. Interestingly, expressing *IDI1-HlPT1L*$_{Δ1-86}$ gene, encoding a fusion enzyme of IPP isomerase and truncated PTase, improved DMX production by 21-fold compared with when *HlPT1L*$_{Δ1-86}$ gene was expressed. This modification could not only increase the supply of DMAPP but also facilitate the contact between DMAPP and PTase with a shortened distance. Therefore, the key limitation for a higher product yield in our studies was the availability of the prenyl donor and colocalization of the substrates and PTase, as supported by the findings of several other studies on the production of prenylated compounds in yeast[38,54,55,57]. In addition, the rate-limiting PTase can be modified by protein engineering to increase its selectivity to specific substrate donors[39,58].

Finally, overexpressing the optimized O-methyltransferase gene *HlOMT1* achieved the de novo biosynthesis of xanthohumol in yeast. However, the low titer (0.14 mg/L) suggested that the activity of HlOMT1 and/or the recycling of SAM should be enhanced to improve the transformation of DMX to xanthohumol, since much more DMX remained in the engineered strain. Furthermore, xanthohumol biosynthesis involves parallel biosynthetic pathways for three precursors *p*-CA-CoA, malonyl-CoA and DMAPP, which is quite different from single-channel biosynthetic pathways such as terpenoid biosynthesis (Supplementary Fig. 1), and thus requires carefully balancing of the parallel biosynthetic modules. In detail, we increased the biosynthesis of precursor *p*-CA-CoA from the aromatic acid pathway by eliminating feedback inhibition and knocking out the competitive pathway. It is challenge to enhance the supply of DMAPP, since its biosynthesis

shares the key node acetyl-CoA with malonyl-CoA and it can be easily converted to FPP. We carefully balanced the pathway flux between providing sufficient prenyl moiety DMAPP and sufficient malonyl moiety malonyl-CoA by tuning the gene expression with promoter replacement, mutating the key enzymes and overexpression of rate-limiting enzymes. We observed that enhancing DMAPP availability through *ERG20* mutation retarded cell growth, which thus requires precise regulation strategies to balance cellular fitness and DMAPP accumulation.

In summary, systematically and modularly rewired the yeast cellular metabolism enabled the de novo microbial production of xanthohumol from the inexpensive carbon source glucose. The strategies for balancing the parallel biosynthesis of precursors, might be helpful for improving the production of other complex natural products.

## Methods
### Strains, plasmids, and reagents
*Escherichia coli* DH5α was used for plasmid construction and amplification. *S. cerevisiae* strain SY03 (*MATa, MAL2-8c, SUC2, his3Δ, ura3-52, gal80Δ, XI-5::P*$_{TEF1}$*-Cas9-T*$_{CYC1}$*)* derived from CEN.PK113-11C (*MATa, MAL2-8c, SUC2, his3Δ, ura3-52*) was used as the background strain for strain construction[59]. The flowchart of yeast strain construction is described in Supplementary Fig. 9. The detailed genotypes of the engineered strains and plasmids are listed in Supplementary Data 2 and Supplementary Data 3, respectively. PrimeStar DNA polymerase for gene amplification was purchased from TaKaRa Biotech (Dalian, China), and 2 × Taq Master Mix polymerase for PCR verification and MultiS One Step Cloning Kit for plasmids construction were purchased from Vazyme Biotech (Nanjing, China). DNA gel purification and plasmid extraction kits were supplied by OMEGA Biotech (USA). All primers (Supplementary Data 4) were synthesized at Sangon Biotech (Shanghai, China). Yeast extracts, tryptone, agar powder, peptone and all other chemicals were from Sangon Biotech unless stated otherwise. All chemical standards were purchased from Sigma–Aldrich unless stated otherwise. The DMX analytical standard was synthesized by Yuanye Biotech (Shanghai, China). All codon optimized heterologous genes (Supplementary Data 5) were synthesized by Genewiz. *EfmvaS* (GenBank-KX064238) and *EfmvaE* (GenBank-KX064239) from *E. faecalis* were synthesized by Genewiz.

### Genetic engineering
Gene knockout and integration were conducted by using a CRISPR/Cas9 system[60]. gRNA-expressing plasmids were constructed according to an efficient assembly method[59]. Briefly, the plasmid backbone fragment and two fragments carrying specific 20 bp sequences were amplified, and then these three fragments were assembled into plasmids in vitro by the MultiS One Step Cloning Kit. Subsequently, the assembled mixes were transformed into *E. coli* and verified by DNA sequencing to obtain the correct plasmid. The specific 20 bp sequence of the gRNA plasmid was designed by the CHOPCHOP webtool (http://chopchop.cbu.uib.no). All donor DNAs for gene deletion and integration were assembled by one-pot fusion PCR and then integrated into the corresponding genome loci[22]. The donor DNAs for gene knockout were prepared by fusing the upstream and downstream homologous arms. The donor DNAs for genome integration were assembled by fusing promoters, target genes, terminators and homologous arms (Supplementary Fig. 10). In situ site-directed mutation of *ERG20* to *ERG20*$^{N127W}$ was conducted according to a two-end selection marker (Two-ESM) method[59]. The mutated *ARO4*$^{K229L}$, *ARO7*$^{G141S}$ and *HMG2*$^{K6R}$ genes were performed by target mutation PCR. The promoter of *ACC1* (from −547 bp to −1 bp) was replaced with P$_{TEF1}$. The promoter of *ERG20*$^{N127W}$ (from −563 bp to −1 bp) was replaced with P$_{HXT1}$ or P$_{ERG1}$.

For enzyme screening, PTase genes and truncated versions (Supplementary Fig. 3) from different organisms were codon-optimized for *S. cerevisiae* and then cloned into pESC-URA with BamHI/HindIII digestion. The *ScCHK* + *IPK*$_{Sc}$ + *HlPT1L*$_{Δ1-86}$ and *EfmvaE*$_{per}$ + *ERG12*$_{per}$ + *EfmvaS*$_{per}$ + *HlPT1L*$_{Δ1-86per}$ fragments were also assembled into pESC-URA and the constructed plasmids were named pESC-IUP and pESC-per, respectively. The IDI1-HlPT1L$_{Δ1-86}$ and CHIL2-HlPT1L$_{Δ1-86}$ fusions were assembled by using a (GGGS)$_3$ linker. Peroxisomal targeting of proteins was ensured by C-terminal addition of peroxisomal signal with the flexible linker GGGS. All transformants of *S. cerevisiae* were verified by colony PCR and DNA sequencing.

Transformation of *E. coli* was performed according to a chemical transformation method[61]. Briefly, the assembled DNA vectors could be directly added to the competent *E. coli* cell suspension. And then, the cell suspension was incubated in an ice bath, heat-pulsed at 42 °C, preincubated at 37 °C with Luria-Bertani (LB) liquid media, poured on LB agar plate and cultured over night at 37 °C.

### Strain cultivation
Unless otherwise specified, *E. coli* strains were grown in Luria-Bertani (LB) medium (10 g/L tryptone, 5 g/L yeast extract, 10 g/L NaCl) at 37 °C and 220 rpm (Zhichu Shaker ZQZY-CS8). In addition, 100 mg/L ampicillin was normally supplemented for plasmid maintenance. Yeast strains were generally cultivated in YPD media consisting of 20 g/L peptone, 10 g/L yeast extract and 20 g/L glucose. Strains containing *URA3* based plasmids were selected on synthetic complete media without uracil (SD-URA), which consisted of 6.7 g/L yeast nitrogen base (YNB) without amino acids and 20 g/L glucose. The *URA3* marker was removed on SD + 5FOA plates containing 6.7 g/L YNB, 20 g/L glucose and 1 g/L 5-fluoroorotic acid (5-FOA). Shake flask batch fermentations were conducted in 100 mL shake flasks with 20 mL of minimal medium (Delft-D) containing 2.5 g/L (NH$_4$)$_2$SO$_4$, 14.4 g/L KH$_2$PO$_4$, 0.5 g/L MgSO$_4$·7H$_2$O, 20 g/L glucose, and trace metal and vitamin solutions[62]. All the above media were supplemented with 40 mg/L histidine and/or 60 mg/L uracil if needed. Then, 20 g/L agar was added to make solid media. The yeast cells were cultivated at 30 °C and 220 rpm in liquid media for 96 h (Zhichu Shaker ZQZY-CS8) with an initial inoculation OD$_{600}$ of 0.2. The maximum specific growth rates (μ$_{max}$) in shake-flask cultivations were determined by the linear regression slope of the natural logarithm of the OD$_{600}$ values versus time curve during the exponential growth phase.

### Product extraction and quantification
For the extraction of xanthohumol and DMX, a low temperature, ultra-high pressure continuous flow cell disrupter was used to disrupt the yeast cells. Ten milliliters of culture broth from shake flask batch fermentation were cyclically broken three times at 1800 MPa. Then, 2 mL of each treated cell culture was added to an equal volume of ethyl acetate and vortexed thoroughly at 1600 rpm for 15 min. The ethyl acetate phase was collected, dried and resuspended in methanol. Before analysis, the extract solution was filtered through a 0.2-μm organic membrane. All extracted samples were quantified by high-performance liquid chromatography (HPLC). Samples were analyzed with a Poroshell 120 EC-C18 column (2.7 μm, 3 × 100 mm, Agilent) on a 1260 infinity II HPLC (Agilent) equipped with a photodiode array detector. Samples were eluted by a gradient method with two solvents: 0.05% formic acid (A) and acetonitrile with 0.05% formic acid (B). The gradient elution conditions were set as follows: 0–10 min, a linear gradient from 20% B to 55% B; 10–20 min, a linear gradient from 55% B to 65% B; 20–23 min, a linear gradient from 65% B to 90% B; 23–24 min, 90% B; 24–26 min, a linear gradient from 90% B to 20% B; then the system was equilibrated using the initial conditions (20% B) for 5 min before the next sample injection. The flow rate was 0.30 mL/min. The target products *p*-CA and N were detected by measuring the absorbance at 288 nm. DMX and xanthohumol were detected by measuring the absorbance at 370 nm. An Agilent 1290 Infinity II UHPLC system coupled to a 6470 A triple quadrupole mass

spectrometer and a ThermoFisher Q Exactive Hybrid Quadrupole-Orbitrap Mass Spectrometer in positive heated electrospray ionization mode was used quantitatively analyze xanthohumol.

## Reporting summary

Further information on research design is available in the Nature Portfolio Reporting Summary linked to this article.

## Data availability

The data supporting the findings of this work are available within the paper and the Supplementary Information files. A reporting summary for this article is available as a Supplementary Information file. Source data are provided with this paper.

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

## Acknowledgements

This study was financially supported by the National Natural Science Foundation of China (21922812) and Key project at central government level: The ability establishment of sustainable use for valuable Chinese medicine resources (2060302). We thank Prof. Jungui Dai (Institute of Materia Medica, Chinese Academy of Medical Sciences) for kindly pro-viding prenyltransferase MaIDT, CtIDT and GuILDT.

## Author contributions

S.Y. and Y.J.Z. conceived the study. S.Y. designed and performed most of the experiments. R.C. contributed to strain construction. X.C. ana-lyzed the data and revised the manuscript. G.W. contributed exogenous genes and analyzed the data. S.Y. and Y.J.Z. wrote the manuscript.

## Competing interests

A patent application (application no. 202310957926.9), for protecting the production of xanthohumol from glucose in *S. cerevisiae*, has been filed by the Dalian Institute of Chemical Physics, Chinese Academy of Sci-ences, with Y.J.Z. and S.Y. named as inventors. Other authors claim no competing interests.
