## [Peer Review File · Nature Communications]

De novo biosynthesis of the hops bioactive flavonoid xanthohumol in yeastReviewers' Comments:

Reviewer #1:

Remarks to the Author:

The author reconstituted the de novo biosynthesis of xanthohumol in baker's yeast and used a set of engineering strategies to optimize the titer of xanthohumol's precursor demethylxanthohumol by 83-fold. Engineering strategies include classic "Push-pull-block" strategy to tune the central metabolism, engineering on bottleneck enzyme PTAs (try different variants, test the optimal pH, protein engineering through truncation), fusion of enzymes to achieve substrate channeling, and tuning the expression levels of pathway enzymes, relocalization of the biosynthesis to peroxisome. The work is solid and the paper is well organized and presented. This work would fit very well in a specific journal, but I am uncertain whether this work aligns with the scope of Nature Communications, given its emphasis on reaching a broad audience.

Reviewer #2:

Remarks to the Author:

This manuscript describes the engineering of baker's yeast to produce the hops natural product, xanthohumol at a titer of ca. 100 ug/L. This compound is a major flavor component of beer, and thus such a yeast strain could have important applications in the flavor and beverage industries.

This general idea of engineering yeast with hops flavor components was already reported in a Nat Comms 2018 paper by Keasling and coworkers (reference 17 in the manuscript). However, this prior work describes the engineering of geraniol and linalool. Therefore, the study here has substantial novelty. The authors should take a few more sentences in the introduction to describe previous efforts to engineer yeast strains for hops flavor components and make it clear why this work is different. Some discussion on the specific value of xanthohumol vs. geraniol and linalool should be provided.

The authors should provide more detailed characterization (e.g. MS spectra) for more biosynthetic intermediates, particularly DMX, similar to what is shown for xanthohumol in Supplementary Figure 7.

The authors provide titers, as is standard, in ug per liter. I would appreciate it if the authors also reported in the main text the scale at which xanthohumol was actually grown I assume it was a culture volume of less than one liter (e.g. 15 ug/100mL).

In the introduction, the authors specifically mention PhytoMetaSyn. This was an important sequencing initiative, but only one of several. Also, references 19 and 20 were from non-Canadian groups who were presumably not supported by PhytoMetaSyn. I think its best if the authors don't mention any specific initiatives and just highlight the wide availability of plant sequence information.

Reviewer #3:

Remarks to the Author:

Review of "De novo biosynthesis of the hops bioactive flavonoid xanthohumol in yeast" by Yang et al.

In this work, the authors reconstructed a pathway for production of the terpenoid xanthohumol, a prenylated flavonoid which is an important flavour compound in beer, in the yeast *Saccharomyces cerevisiae*. They used a number of strategies to redirect carbon flux into the target product, including "fine-tuning of the competitive metabolic pathways, prenyltransferase engineering, enhancing precursor supply, substrate channeling, and peroxisomal engineering". This led to an >80-fold increase in the xanthohumol precursor demethylxanthohumol (DMX) and established a full pathway to xanthohumol in yeast. This work also identified prenylation as a limiting step in the pathway, allowing

the authors to target DMAPP supply as a key engineering step to improve flux. In addition they used enzyme fusion/scaffolding (referred to incorrectly as 'substrate channelling' by the authors) to redirect carbon flux to DMX.

Commentary:

1. Each organism has a native ratio of the universal prenylated terpenoid precursors DMAPP and IPP; this ratio is controlled by the organism's HDR enzyme, and has likely evolved to support that organism's unique need for short- vs. longer-chain terpenoids (Bongers et al., eLife 2020;9:e48685). Production of novel prenylated products in yeast may result in disturbance of this balance and subsequent limitation of a prenylated precursor, which may either limit titres or result in metabolic imbalance with potential growth defects. As noted by the authors, balancing of the prenylated precursors has the potential to be a universal metabolic engineering approach for production of prenylated products in yeast, in particular, where the prenyl precursor (DMAPP in this case) is limiting. To my knowledge this is a novel observation with respect to the metabolic engineering of prenylated products, and as noted will be generalisable. This point is, I believe, the crux of the novelty of the work, and could be better highlighted in the Abstract and Discussion to highlight the impact of the work. The abstract is currently focussed on the novelty of establishing the pathway to xanthohumol, but establishment of new natural product pathways is pretty plentiful in the literature and the other metabolic engineering approaches deployed in this work are all well-established. Combining established approaches is not particularly novel.
2. Another aspect of this work that I find impressive is the tour-de-force of modifying three different precursor pathways required to make the complex prenylated flavonoid and the subsequent necessary investigations to identify and mitigate the rate limiting steps. The authors should be congratulated for the sheer amount of work that has gone into this manuscript.
3. While the final titres are not particularly high (in the mg range), the authors have nonetheless delivered an impressive piece of work with substantial titre increases and with novel and complex metabolic engineering strategies employed. Moreover, as noted, their findings around how to balance pathway flux between providing sufficient prenyl moiety and sufficient malonyl moiety is the novelty of the work. The authors need to develop this component of the work much better.

Detailed comments

General:

4. The English language is very good considering that the authors presumably all have English as a second language, congratulations on this nice work. There are just a few places that need correction, and this should be picked up by the copyeditor.

Abstract:

5. it might be worth noting that climate change is starting to affect hops production, and will likely result in further production restrictions in the future
6. line 53: first mention of nutraceutical applications. What are the applications for xanthohumol?
7. Lines 58-60: it's strange to mention a national initiative and a specific project that failed to produce the target product in a manuscript. Suggest removing this line as the people in the project may find it offensive to be singled out as failures! Or perhaps re-word to soften the phrasing a bit, for example, the authors could simply note that the commercial value of xanthohumol has been identified in previous programs (18), but has not yet successfully been produced.

Results and Discussion

8. tHMG1+IDI1 had no discernible effect on N and NC production. Failure of this step to deliver results would suggest prima facie that the mevalonate pathway was not rate-limiting – similarly to the authors' conclusion from the ACC1 engineering step. Could the authors please explain their reasoning for further investigating at this point? There is no evidence at this stage of the work that the tHMG1+IDI1 actually worked – the next normal step would be to confirm that it worked by, for example, introducing a sesquiterpene synthase to the tHMG1+IDI1 strain (YS112) and the strain

immediately previous to this strain (YS111). If sesquiterpene production increased in the absence of an increase in NC production, it would then suggest a limitation in DMAPP or prenyltransferase activity. This is a logical leap in the manuscript narrative that should be closed.

9. FPPS is indeed known to be a very efficient enzyme, and it is well documented in the literature that it competes effectively with monoterpene synthases for the GPP intermediate during FPP biosynthesis. So targeting it to increase DMAPP makes sense. However, I am intrigued by the result of the FPP(N127W) mutation, which improved DMX and NC/N production by significantly. The authors suggest that this 'might be due to driving the metabolic flux toward L-tyrosine and malonyl-CoA from the downregulation of FPP biosynthesis'. If find it to be very unlikely that tyrosine production is affected by a change in FPPS activity because of the metabolic distance between the MVA pathway and the tyrosine. It's much more likely that there is an effect on malonyl-CoA, because MVA production is initiated from acetyl-CoA and is consequently relatively close in the metabolic pathway map. However, the authors have just suggested that this pathway is not limiting, because ACC1 engineering is not effective for increasing DMX and NC/N. This result requires a more nuanced explanation than the one offered by the authors. In fact, this is a key finding: dealing with the DMAPP availability by downregulating the FPPS activity results in both increased availability of DMAPP and re-balancing of flux towards malonyl-CoA, which delivers improved production of target compounds that have both malonyl-CoA and prenyl (DMAPP) moieties involved in their biosynthesis. This hypothesis is supported by the authors subsequent finding, that pushing harder into the MVA pathway decreased DMX production, presumably because of reduced malonyl-CoA availability due to competition from the MVA pathway for precursors. Again this could be discussed in a more nuanced way, and I believe that this is the most interesting finding of the work.

10. Lines 120-121 'to improve DMAPP accumulation' – authors have not measured DMAPP so cannot make this statement. In fact, it's more likely that they improved DMAPP availability for prenylation – not 'accumulation' in the cell. Change to 'to improve DMAPP availability'

11. Growth curves and specific growth rate should be presented in Figure 2, and in all figures with engineering steps, because (a) it's important to understand if any modifications cause a growth or biomass yield penalty, and (b) the primary product of the MVA in terms of carbon yield is ergosterol, which is a significant biomass component

12. Add NC and N data to Figure 2b for strain YS117

13. note that the Keasling group has identified prenylation as the rate-limiting step in cannabinoid production in cell factories, similarly to here – this should be included in the discussion. There's a German group which may have also published on prenylation as rate-limiting in cannabinoid production, the name escapes me but it should be easy to find.

14. The authors really hammered on the PTase to improve it, without demonstrating good evidence that that step was the sole limiting step (indeed, their earlier experiment indicates that the malonyl-CoA step (most likely – see comments above) was limiting and was de-bottlenecked by decreasing FPPS activity. While the approach of hammering on the prenyltransferase was successful, it would be good to see the justification behind this approach better explained. Referring to previous findings that have identified the prenyltransferase step as limiting (e.g. Keasling's monoterpene prenyltransferase cannabinoid work) might be helpful, but perhaps there are other reasons that the authors pursued this approach?

15. Lines 207-208: 'At the same time, DMAPP was dynamically present in cells, as the accumulation of DMAPP has been reported to be toxic to host cells^{48, 49}'. Review these references carefully and you will likely observe that the authors did not measure DMAPP (or any other prenyl phosphates) and that toxicity is an assumption based on the engineering results. Moreover, there is considerable discussion in the community regarding which (if any) prenyl phosphate compound is toxic (DMAPP? IPP? GPP? FPP?). I do not believe that this question has been adequately resolved in the literature and thus is remains speculative. Moreover, stating that 'DMAPP was dynamically present in the cells' is also incorrect, unless this can be proven by measuring the DMAPP. While possible, it is a significant technical challenge to do this, requiring advanced analytics capabilities owing to the similarity between IPP and DMAPP and the difficulty in separating them (it's more feasible to measure both and make a statement about the general accumulation of C5 prenyl diphosphates).

16. Please do not refer to scaffolding of IDI and DMAPP as 'substrate channelling' anywhere in the

manuscript. Substrate channelling is a specific and evolved trait between enzymes that have co-evolved to pass the substrate directly from one active site to another. Crudely scaffolding enzymes together like we do in metabolic engineering does not achieve this. It simply increases the local concentration of the substrate for the second enzyme in the catalytic cascade, providing it a better opportunity to grab the substrate as it emerges from the previous enzyme and improved local catalytic conditions. It's just scaffolding.

17. There might already be a partial MVA pathway for DMAPP production in the peroxisome, which might explain why the addition of a DMAPP production pathway didn't help in the initial stage, whereas overexpressing an upstream MVA pathway to increase precursor availability did help.

18. There is cross-talk between compartments, likely at the prenyl phosphate level, but also at other levels (e.g. MVA). This may be necessary to consider when analysing and discussing the results.

19. There's a lot of repetition between the Results and Discussion. Consider combining these sections if the journal format allows combined Results and Discussion. Co-locating the deeper discussion from the Discussion section with data and preliminary discussion in the Results would improve the readability of the manuscript.

20. The final paragraph is a bit weak. It describes what was done, but does not describe the broader impact of the work. Consider the above commentary about where the real novelty of this work is to help craft a more impactful final paragraph.

Figures

21. Figure 1: $\text{DMAPP} + 2 \text{IPP} \rightleftharpoons \text{FPP} + 2 \text{Pi}$. Correct on the figure to show the condensation reactions that make FPP

22. Figure 1: IDI1 is an isomerase, the arrows should point both ways. The same in figure 2a.

Claudia Vickers
Queensland University of Technology
BioBuilt Solutions

Responses to the reviewers' comments (NCOMMS-23-22890-T)

Dear Reviewers

We appreciate all your constructive comments and suggestions, which definitely gave us helpful guidance to make our manuscript stronger. We have revised our manuscript accordingly, and the changes in the revised text are highlighted in red. Besides this, we carefully checked the whole manuscript and corrected other typos. All our responses to your comments are listed as follows:

Responses to Reviewer 1

Reviewer #1 (Remarks to the Author):

The author reconstituted the de novo biosynthesis of xanthohumol in baker's yeast and used a set of engineering strategies to optimize the titer of xanthohumol's precursor demethylxanthohumol by 83-fold. Engineering strategies include classic "Push-pull-block" strategy to tune the central metabolism, engineering on bottleneck enzyme PTas (try different variants, test the optimal pH, protein engineering through truncation), fusion of enzymes to achieve substrate channeling, and tuning the expression levels of pathway enzymes, relocalization of the biosynthesis to peroxime. The work is solid and the paper is well organized and presented. This work would fit very well in a specific journal, but I am uncertain whether this work aligns with the scope of Nature Communications, given its emphasis on reaching a broad audience.

Reply: We appreciate your positive evaluation on our work. Engineering microbial production is considered a feasible approach to sustainable supply of low content natural productions without the dependence of land and specific cultural conditions. We thus think our study is common interest to a broad audience. For examples, Denby et al engineered yeast for production of flavor monoterpenes (Nat Commun, 2018, 9, 965) and Liu et al engineered yeast for overproduction of isoflavonoids (Nat Commun, 2021, 12, 6085).

Responses to Reviewer 2

Reviewer #2 (Remarks to the Author):

This manuscript describes the engineering of baker's yeast to produce the hops natural product, xanthohumol at a titer of ca. 100 ug/L. This compound is a major flavor component of beer, and thus such a yeast strain could have important applications in the flavor and beverage industries.

Reply: Thanks for your positive comments on our manuscript, and we provide a point-by-point reply to address your concerns.

This general idea of engineering yeast with hops flavor components was already reported in a Nat Comms 2018 paper by Keasling and coworkers (reference 17 in the manuscript). However, this prior work describes the engineering of geraniol and linalool. Therefore, the study here has substantial novelty. The authors should take a few more sentences in the introduction to describe previous efforts to engineer yeast strains for hops flavor components and make it clear why this work is different. Some discussion on the specific value of xanthohumol vs. geraniol and linalool should be provided.

Reply: Thanks for your suggestions. We have added more details to emphasize our study in the revised manuscript. Thus, engineering *S. cerevisiae* for such biosyntheses might provide a sustainable route to supply large amounts of xanthohumol for potential applications in pharmaceuticals and enhance the xanthohumol level in beer, since engineering biosynthesis of monoterpene linalool and geraniol in brewing yeast gives rise to a hoppy beer flavor (Ref. 17). In particular, engineering xanthohumol in brewing yeast not only give the full flavor of beer, and also provide sufficient supply of xanthohumol for functional food and pharmaceuticals (Line 55-61).

The authors should provide more detailed characterization (e.g. MS spectra) for more biosynthetic intermediates, particularly DMX, similar to what is shown for xanthohumol in Supplementary Figure 7.

Reply: Related data has been supplemented in the revised manuscript. As shown in revised Supplementary Fig. 7c, the intermediate DMX of strain YSC3 was analyzed by

LC/MS-MS.

The authors provide titers, as is standard, in ug per liter. I would appreciate it if the authors also reported in the main text the scale at which xanthohumol was actually grown I assume it was a culture volume of less than one liter (e.g. 15 ug/100mL).

Reply: Thanks for your advice, and all strains in manuscript were fermented in 100 mL shake flasks with 20 mL of minimal medium, which further was emphasized in the methods and materials section. And we use the normal unified mg/L and $\mu\text{g/L}$ in the whole manuscript.

In the introduction, the authors specifically mention PhytoMetaSyn. This was an important sequencing initiative, but only one of several. Also, references 19 and 20 were from non-Canadian groups who were presumably not supported by PhytoMetaSyn. I think its best if the authors don't mention any specific initiatives and just highlight the wide availability of plant sequence information.

Reply: Thanks for your critical comments, and we agree with your argument, so we removed related description in the revised manuscript.

Responses to Reviewer 3

Reviewer #3 (Remarks to the Author):

Review of “De novo biosynthesis of the hops bioactive flavonoid xanthohumol in yeast” by Yang et al.

In this work, the authors reconstructed a pathway for production of the terpenoid xanthohumol, an prenylated flavonoid which is an important flavour compound in beer, in the yeast *Saccharomyces cerevisiae*. They used a number of strategies to redirect carbon flux into the target product, including ‘fine-tuning of the competitive metabolic pathways, prenyltransferase engineering, enhancing precursor supply, substrate channeling, and peroxisomal engineering’. This led to an >80-fold increase in the xanthohumol precursor demethylxanthohumol (DMX) and established a full pathway to xanthohumol in yeast. This work also identified prenylation as a limiting step in the pathway, allowing the authors to target DMAPP supply as a key engineering step to improve flux. In addition they used enzyme fusion/scaffolding (referred to incorrectly as ‘substrate channelling’ by the authors) to redirect carbon flux to DMX.

Reply: We thank your positive evaluation on our work. More importantly, we really appreciate your helpful comments, in particular the DMAPP availability analysis that help to strengthen our manuscript. We here provide a point-by-point reply to address all your concerns. In addition, sorry for our confusing description on enzyme fusion, and we carefully revised these descriptions in the revised manuscript.

Commentary:

1. Each organism has a native ratio of the universal prenylated terpenoid precursors DMAPP and IPP; this ratio is controlled by the organism’s HDR enzyme, and has likely evolved to support that organism’s unique need for short- vs. longer-chain terpenoids (Bongers et al., eLife 2020;9:e48685). Production of novel prenylated products in yeast may result in disturbance of this balance and subsequent limitation of a prenylated precursor, which may either limit titres or result in metabolic imbalance with potential growth defects. As noted by the authors, balancing of the prenylated precursors has the potential to be a universal metabolic engineering approach for production of prenylated

products in yeast, in particular, where the prenyl precursor (DMAPP in this case) is limiting. To my knowledge this is a novel observation with respect to the metabolic engineering of prenylated products, and as noted will be generalisable. This point is, I believe, the crux of the novelty of the work, and could be better highlighted in the Abstract and Discussion to highlight the impact of the work. The abstract is currently focussed on the novelty of establishing the pathway to xanthohumol, but establishment of new natural product pathways is pretty plentiful in the literature and the other metabolic engineering approaches deployed in this work are all well-established. Combining established approaches is not particularly novel.

Reply: Thanks for your suggestive comment. As you suggested, we highlight our novelty on engineering the supply prenyl precursor DMAPP in Abstract (Line 32-36) and Discussion (Line 301-305).

2. Another aspect of this work that I find impressive is the tour-de-force of modifying three different precursor pathways required to make the complex prenylated flavonoid and the subsequent necessary investigations to identify and mitigate the rate limiting steps. The authors should be congratulated for the sheer amount of work that has gone into this manuscript.

Reply: We appreciate your encouraged comments on our work.

3. While the final titres are not particularly high (in the mg range), the authors have nonetheless delivered an impressive piece of work with substantial titre increases and with novel and complex metabolic engineering strategies employed. Moreover, as noted, their findings around how to balance pathway flux between providing sufficient prenyl moiety and sufficient malonyl moiety is the novelty of the work. The authors need to develop this component of the work much better.

Reply: We appreciate your positive evaluation on our work.

Detailed comments

General:

4. The English language is very good considering that the authors presumably all have English as a second language, congratulations on this nice work. There are just a few places that need correction, and this should be picked up by the copyeditor.

Reply: We thank your positive evaluation on our work. And we revised the manuscript carefully.

Abstract:

5. it might be worth noting that climate change is starting to affect hops production, and will likely result in further production restrictions in the future

Reply: Related information was added in Abstract (Line 21-22).

6. line 53: first mention of nutraceutical applications. What are the applications for xanthohumol?

Reply: Sorry for the confusing description. Xanthohumol is still in the medical research stage and is not yet available on the market. And we modified this sentence.

7. Lines 58-60: it's strange to mention a national initiative and a specific project that failed to produce the target product in a manuscript. Suggest removing this line as the people in the project may find it offensive to be singled out as failures! Or perhaps reword to soften the phrasing a bit, for example, the authors could simply note that the commercial value of xanthohumol has been identified in previous programs (18), but has not yet successfully been produced.

Reply: Thanks for reminding us, and we agree with your argument, so we removed related description in the revised manuscript.

Results and Discussion

8. tHMG1+IDI1 had no discernible effect on N and NC production. Failure of this step to deliver results would suggest prima facie that the mevalonate pathway was not rate-limiting – similarly to the authors' conclusion from the ACC1 engineering step. Could the authors please explain their reasoning for further investigating at this point? There is no evidence at this stage of the work that the tHMG1+IDI1 actually worked – the next normal step would be to confirm that it worked by, for example, introducing a sesquiterpene synthase to the tHMG1+IDI1 strain (YS112) and the strain immediately previous to this strain (YS111). If sesquiterpene production increased in the absence of an increase in NC production, it would then suggest a limitation in DMAPP or prenyltransferase activity. This is a logical leap in the manuscript narrative that should be closed.

Reply: Thanks for your suggestive comments and sorry for the unclear descriptions. DMAPP and NC are the precursors for DMX production that catalyzed by PTase (Fig. R3). Since *tHMG1*+*IDI1* was considered as the two limiting steps in MVA pathway and we previously showed overexpression of *tHMG1* and *IDI1* improved (+)-Valencene production in *S. cerevisiae* (J. Agric. Food Chem. 2022, 70, 7180–7187), which suggested that *tHMG1* and *IDI1* were bottleneck for DMAPP supply. The failure of DMAPP production in strain YS112 with overexpressing *tHMG1* and *IDI1* might be attributed to the draining DMAPP toward FPP biosynthesis. Consistently, replacing the FPP synthase mutation *ERG20*^{N127W} with decreased activity in catalyzing DMAPP turnover (Ref. 27), succeeded in producing 12 µg/L DMX in strain YS116. The detailed description can be found in line 114-122, and more detailed explanation was added in line 123-124.

Fig. R3. The DMAPP availability is essential for DMX biosynthesis.

9. FPPS is indeed known to be a very efficient enzyme, and it is well documented in the literature that it competes effectively with monoterpene synthases for the GPP intermediate during FPP biosynthesis. So targeting it to increase DMAPP makes sense. However, I am intrigued by the result of the FPP(N127W) mutation, which improved DMX and NC/N production by significantly. The authors suggest that this ‘might be due to driving the metabolic flux toward L-tyrosine and malonyl-CoA from the downregulation of FPP biosynthesis’. If find it to be very unlikely that tyrosine production is affected by a change in FPPS activity because of the metabolic distance between the MVA pathway and the tyrosine. It’s much more likely that there is an effect

on malonyl-CoA, because MVA production is initiated from acetyl-CoA and is consequently relatively close in the metabolic pathway map. However, the authors have just suggested that this pathway is not limiting, because ACC1 engineering is not effective for increasing DMX and NC/NF. This result requires a more nuanced explanation than the one offered by the authors. In fact, this is a key finding: dealing with the DMAPP availability by downregulating the FPPS activity results in both increased availability of DMAPP and re-balancing of flux towards malonyl-CoA, which delivers improved production of target compounds that have both malonyl-CoA and prenyl (DMAPP) moieties involved in their biosynthesis. This hypothesis is supported by the authors subsequent finding, that pushing harder into the MVA pathway decreased DMX production, presumably because of reduced malonyl-CoA availability due to competition from the MVA pathway for precursors. Again this could be discussed in a more nuanced way, and I believe that this is the most interesting finding of the work.

Reply: Thanks for your suggestive comments, which enlightened us a lot. Actually, ACC1 engineering (replacing the ACC1 promoter by TEF1 promoter, strain YS111) simply changed the metabolic flux from acetyl-CoA to malonyl-CoA. However, reducing the activity of FPPS could not only increase the availability of DMAPP, but also increase the acetyl-CoA pool to malonyl-CoA. Therefore, our subsequent modification of the MVA pathway further affected the flux direction of the acetyl-CoA pool, thus affecting the availability of malonyl-CoA. Related explanation was also involved in the revised manuscript (Line 119-124 and Line 186-191)

10. Lines 120-121 ‘to improve DMAPP accumulation’ – authors have not measured DMAPP so cannot make this statement. In fact, it’s more likely that they improved DMAPP availability for prenylation – not ‘accumulation’ in the cell. Change to ‘to improve DMAPP availability’

Reply: We corrected this description in the revised manuscript.

11. Growth curves and specific growth rate should be presented in Figure 2, and in all figures with engineering steps, because (a) it’s important to understand if any modifications cause a growth or biomass yield penalty, and (b) the primary product of

the MVA in terms of carbon yield is ergosterol, which is a significant biomass component

Reply: Thanks for reminding us. The final OD₆₀₀ of strains was added in the revised Fig. 2b and the revised Supplementary Fig. 3. We can see that some metabolic modifications retarded cell growth. In particular, the mutation of FPPS significantly reduced final OD₆₀₀ which might be due to the affect the biosynthesis significant biomass component ergosterol, as you suggested. However, overexpression of rate-limiting enzymes of the MVA pathway had little effect on cell growth (revised Supplementary Fig. 3b). Other modified strains had little effect on cell growth, so the final OD₆₀₀ of these strains was not presented in the revised manuscript.

12. Add NC and N data to Figure 2b for strain YS117

Reply: Related data was supplemented in the revised Fig. 2b.

13. note that the Keasling group has identified prenylation as the rate-limiting step in cannabinoid production in cell factories, similarly to here – this should be included in the discussion. There's a Germen group which may have also published on prenylation as rate-limiting in cannabinoid production, the name escapes me but it should be easy to find.

Reply: Thanks for your kindly clarification, and we added related information in Discussion: Previous studies also showed that the prenylation was a limiting step in cannabinoid biosynthesis (Line 296-297). And the related papers were cited at here (J. Biotechnol., 2017, 259, 204–212; Nature, 2019, 567, 7746).

14. The authors really hammered on the PTase to improve it, without demonstrating good evidence that that step was the sole limiting step (indeed, their earlier experiment indicates that the malonyl-CoA step (most likely – see comments above) was limiting and was de-bottlenecked by decreasing FPPS activity. While the approach of hammering on the prenyltransferase was successful, it would be good to see the justification behind this approach better explained. Referring to previous findings that have identified the prenyltransferase step as limiting (e.g. Keasling's monoterpene prenyltransferase cannabinoid work) might be helpful, but perhaps there are other

reasons that the authors pursued this approach?

Reply: Thanks for your suggestions. As your last comments, previous studies showed that the prenylation was a limiting step in cannabinoid biosynthesis (J. Biotechnol., 2017, 259, 204–212; Nature, 2019, 567, 7746). In our study, the DMX production was much lower than the precursor NC production, which suggested the prenylation was a bottleneck in DMX biosynthesis. Further analysis could be found in Line 314-318 of Discussion section. In addition, we added the related discussion in the revised manuscript (Line 294-297, Line 301-305).

15. Lines 207-208: ‘At the same time, DMAPP was dynamically present in cells, as the accumulation of DMAPP has been reported to be toxic to host cells^{48, 49}’. Review these references carefully and you will likely observe that the authors did not measure DMAPP (or any other prenyl phosphates) and that toxicity is an assumption based on the engineering results. Moreover, there is considerable discussion in the community regarding which (if any) prenyl phosphate compound is toxic (DMAPP? IPP? GPP? FPP?). I do not believe that this question has been adequately resolved in the literature and thus it remains speculative. Moreover, stating that ‘DMAPP was dynamically present in the cells’ is also incorrect, unless this can be proven by measuring the DMAPP. While possible, it is a significant technical challenge to do this, requiring advanced analytics capabilities owing to the similarity between IPP and DMAPP and the difficulty in separating them (it’s more feasible to measure both and make a statement about the general accumulation of C5 prenyl diphosphates).

Reply: Yes, you are completely right, we removed the description and modified the sentence: Since *IDI1* catalyzed the isomerization of IPP into DMAPP, we attempted to fuse *IDI1* with *HIPTIL*_{Δ1-83} to help the quick interaction of DMAPP with PTase.

16. Please do not refer to scaffolding of *IDI* and DMAPP as ‘substrate channelling’ anywhere in the manuscript. Substrate channelling is a specific and evolved trait between enzymes that have co-evolved to pass the substrate directly from one active site to another. Crudely scaffolding enzymes together like we do in metabolic engineering does not achieve this. It simply increases the local concentration of the

substrate for the second enzyme in the catalytic cascade, providing it a better opportunity to grab the substrate as it emerges from the previous enzyme and improved local catalytic conditions. It's just scaffolding.

Reply: Thanks for your advice, and we agree with your argument. We revised it in the revised manuscript.

17. There might already be a partial MVA pathway for DMAPP production in the peroxisome, which might explain why the addition of a DMAPP production pathway didn't help in the initial stage, whereas overexpressing an upstream MVA pathway to increase precursor availability did help.

Reply: You are definitely right. The peroxisome has a single-layer membrane that allows a large number of low molecular weight compounds to travel across, so IPP/DMAPP can enter the peroxisome from the cytosol. However, targeting the complete exogenous pathway and IDI1 to the peroxisome did not obtain the target product DMX (strain YS2per in the revised Fig.5b), indicating that the availability of DMAPP in the peroxisome was not enough. And reconstruction of the complete MVA pathway (strain YS5per in the revised Fig.5c) and overexpression of the upstream MVA pathway (strains YS11per and YS12per in the revised Fig.5d) in the peroxisome improved DMX production, which suggested that that overexpression of the MVA pathway in peroxisome could improve the availability of DMAPP. The related description can be found in line 259-265.

18. There is cross-talk between compartments, likely at the prenyl phosphate level, but also at other levels (e.g. MVA). This may be necessary to consider when analysing and discussing the results.

Reply: Thanks for your suggestions. It is worthy to investigate the cross-talk between compartments for DMAPP supply and NC biosynthesis, which is however very complex and lacks of sufficient evidence in this study. We thus added some assumption in our revised manuscript (Line 269-270).

19. There's a lot of repetition between the Results and Discussion. Consider combining these sections if the journal format allows combined Results and Discussion. Co-locating

the deeper discussion from the Discussion section with data and preliminary discussion in the Results would improve the readability of the manuscript.

Reply: We carefully revised the Discussion section to avoid the repetition and made more deep analysis in the Discussion.

20. The final paragraph is a bit weak. It describes what was done, but does not describe the broader impact of the work. Consider the above commentary about where the real novelty of this work is to help craft a more impactful final paragraph.

Reply: We revised the final paragraph to highlight the potential impact of our work.

Figures

21. Figure 1: $DMAPP + 2 IPP \rightarrow FPP + 2 Pi$. Correct on the figure to show the condensation reactions that make FPP

Reply: Thanks for your advice, and we corrected it in the revised figure.

22. Figure 1: IDI1 is an isomerase, the arrows should point both ways. The same in figure 2a.

Reply: Thanks for your advice, and we corrected it in the revised figure.

Reviewers' Comments:

Reviewer #2:

Remarks to the Author:

My comments were largely addressed.

Could I please request that when the titers are mentioned in the main text as mg/L, could the authors please put in parantheses (xxug/20mL). I know this is not standard, but I feel that much of the metabolic engineering literature can be misleading because these titers often do not scale linearly. I make this request of all such papers that I review. Thank you.

Reviewer #3:

Remarks to the Author:

See attached document

The manuscript is much improved. However, a couple of my points from the first review are not well addressed. Specifically (numbering is original question numbering):

3. While the final titres are not particularly high (in the mg range), the authors have nonetheless delivered an impressive piece of work with substantial titre increases and with novel and complex metabolic engineering strategies employed. Moreover, as noted, their findings around how to balance pathway flux between providing sufficient prenyl moiety and sufficient malonyl moiety is the novelty of the work. The authors need to develop this component of the work much better.

Reply: We appreciate your positive evaluation on our work.

Response to reply: The authors have not properly addressed this concern. They have added the following red text to the Abstract and Discussion:

Abstract:

We also revealed that prenylation is the key limiting step in DMX biosynthesis and developed several metabolic regulation strategies to enhance the supply of dimethylallyl pyrophosphate (DMAPP), which should be helpful for regulation the DMAPP availability and improving the production of prenylated natural products.

Discussion:

In addition to PTase activity, the limited availability of DMAPP is another bottleneck for efficient prenylation^{54,55}, since DMAPP level is tightly regulated and efficiently transformed toward FPP in ergosterol biosynthesis in yeast⁵⁶. We here developed several pathway engineering strategies to enhance the DMAPP supply and enzyme fusion improve the DMAPP availability to PTase.

...

(and the added final paragraph):

In summary, systematically and modularly rewired the yeast cellular metabolism enabled the de novo microbial production of xanthohumol from the inexpensive carbon source glucose. The strategies for balancing the parallel biosynthesis of precursors, might be helpful for improving the production of other complex natural products.

However, these points do not clearly highlight the novelty of the work. As noted originally, the novelty of the work lies in the authors findings about how to balance pathway flux between providing sufficient prenyl moiety and sufficient malonyl moiety. To explore this properly in the discussion, the authors need to add more than a sentence referring to DMAPP synthesis and a vague sentence talking about balancing parallel biosynthesis of precursors. The need to discuss where each precursor comes from and why balancing between these pathways might be critical in this case, and also why the technique they applied here might be generalisable for engineering of these kinds of compounds (and possibly other examples). Without a proper exploration of the generalisable nature of the work, it lacks novelty and impact for metabolic engineering.

This also needs to be better explained in the Abstract. The current added sentence does not convey the significance of the work.

11. Growth curves and specific growth rate should be presented in Figure 2, and in all figures with engineering steps, because (a) it's important to understand if any modifications cause a growth or biomass yield penalty, and (b) the primary product of the MVA in terms of carbon yield is ergosterol, which is a significant biomass component

Reply: Thanks for reminding us. The final OD600 of strains was added in the revised Fig. 2b and the revised Supplementary Fig. 3. We can see that some metabolic modifications retarded cell growth. In particular, the mutation of FPPS significantly reduced final OD600 which might be due to the affect the biosynthesis significant biomass component ergosterol, as you suggested. However, overexpression of ratelimiting enzymes of the MVA pathway had little effect on cell growth (revised Supplementary Fig. 3b). Other modified strains had little effect on cell growth, so the final OD600 of these strains was not presented in the revised manuscript.

Response to reply: A final OD reading is not a growth curve, and it is not a specific growth rate. It provides a limited 'snapshot in time' that gives only a small amount of information about the strain's growth behaviour. Final OD very subject to bias from the inoculation conditions and sampling time. Growth curves are not hard to do, and neither is calculation of the specific growth rate. They are important for interpreting your data. You need to understand if they grow at a normal rate but hit a growth bottleneck and plateaux early, or if they grow slowly and reach a lower OD, or if they peak and drop down (indicating cell lysis) – etc. I presume that you don't already have this data and will need to collect it by repeating fermentations; in that case, simply note that separate fermentaitons were used to collect the growth rate data and the metabolite data. That will be fine, as long as you grow the cultures under identical conditions, and the final OD readings are taken at the same time as the final OD readings you already have, and the final OD readings between the two experiments agree with each other.

Moreover, the presentation of the final OD data on Figure 2b and Suppl. 3b is wrong. You have connected the data with a line plot, indicating changes over time or linked changes. This is wrong – you cannot present discrete data points this way. A bar chart (e.g. a second data set on the bar chart you already have) would be appropriate. In addition, you should prepare growth curves (can be provided as supplementary material) and calculate specific growth rate (add to your figures in the table below the other strain data) for ALL strains – even the ones that don't show a difference. Just because there is no difference doesn't mean you don't have to show the data. Finally, you need to properly explore this data and what it means for interpretation of your other data in the Discussion.

There remain several grammar and punctuation errors, but these should be picked up at the copyeditor stage.

Responses to the reviewers' comments (NCOMMS-23-22890A)

Dear Reviewers

We appreciate all your constructive comments and suggestions, which definitely gave us helpful guidance to make our manuscript stronger. We have revised our manuscript accordingly, and the changes in the revised text are highlighted in blue. Besides this, we carefully checked the whole manuscript and corrected other typos. All our responses to your comments are listed as follows:

Reviewer #2 (Remarks to the Author):

My comments were largely addressed.

Could I please request that when the titers are mentioned in the main text as mg/L, could the authors please put in parantheses (xxug/20mL). I know this is not standard, but I feel that much of the metabolic engineering literature can be misleading because these titers often do not scale linearly. I make this request of all such papers that I review. Thank you.

Reply: Thanks for your suggestions. For metabolic engineering, in order to make readers more intuitive to understand the impact of metabolic modification on production, ug/L or mg/L is a normal used unit of concentration in most literatures. But as you said titers often did not scale linearly, so we added a description of fermentation scale to each figure legends according to your comments. All strains were cultivated in 100 mL shake flasks containing 20 mL of minimal medium.

Reviewer #3 (Remarks to the Author):

See attached document

The manuscript is much improved. However, a couple of my points from the first review are not well addressed. Specifically (numbering is original question numbering):

3. While the final titres are not particularly high (in the mg range), the authors have nonetheless delivered an impressive piece of work with substantial titre increases and with novel and complex metabolic engineering strategies employed. Moreover, as noted, their findings around how to balance pathway flux between providing sufficient prenyl moiety and sufficient malonyl moiety is the novelty of the work. The authors need to develop this component of the work much better.

However, these points do not clearly highlight the novelty of the work. As noted originally, the novelty of the work lies in the authors findings about how to balance pathway flux between providing sufficient prenyl moiety and sufficient malonyl moiety.

To explore this properly in the discussion, the authors need to add more than a sentence referring to DMAPP synthesis and a vague sentence talking about balancing parallel biosynthesis of precursors. The need to discuss where each precursor comes from and why balancing between these pathways might be critical in this case, and also why the technique they applied here might be generalisable for engineering of these kinds of compounds (and possibly other examples). Without a proper exploration of the generalisable nature of the work, it lacks novelty and impact for metabolic engineering.

This also needs to be better explained in the Abstract. The current added sentence does not convey the significance of the work.

Reply: Thanks for your suggestive comment. As you suggested, we highlight our novelty on balancing the parallel biosynthetic pathways in Abstract (Line 29-31) and Discussion (Line 328-334).

11. Growth curves and specific growth rate should be presented in Figure 2, and in all figures with engineering steps, because (a) it's important to understand if any modifications cause a growth or biomass yield penalty, and (b) the primary product of

the MVA in terms of carbon yield is ergosterol, which is a significant biomass component

Response to reply: A final OD reading is not a growth curve, and it is not a specific growth rate. It provides a limited 'snapshot in time' that gives only a small amount of information about the strain's growth behaviour. Final OD very subject to bias from the inoculation conditions and sampling time. Growth curves are not hard to do, and neither is calculation of the specific growth rate. They are important for interpreting your data. You need to understand if they grow at a normal rate but hit a growth bottleneck and plateaux early, or if they grow slowly and reach a lower OD, or if they peak and drop down (indicating cell lysis) – etc. I presume that you don't already have this data and will need to collect it by repeating fermentations; in that case, simply note that separate fermentations were used to collect the growth rate data and the metabolite data. That will be fine, as long as you grow the cultures under identical conditions, and the final OD readings are taken at the same time as the final OD readings you already have, and the final OD readings between the two experiments agree with each other.

Moreover, the presentation of the final OD data on Figure 2b and Suppl. 3b is wrong. You have connected the data with a line plot, indicating changes overtime or linked changes. This is wrong – you cannot present discrete data points this way. A bar chart (e.g. a second data set on the bar chart you already have) would be appropriate. In addition, you should prepare growth curves (can be provided as supplementary material) and calculate specific growth rate (add to your figures in the table below the other strain data) for ALL strains – even the ones that don't show a difference. Just because there is no difference doesn't mean you don't have to show the data. Finally, you need to properly explore this data and what it means for interpretation of your other data in the Discussion.

Reply: Thanks for your suggestions. We have corrected the wrong presentation of OD₆₀₀ in the revised Supplementary Fig. 2 and the revised Supplementary Fig. 4. As you suggested, the final OD₆₀₀ value only reflected a part of the growth of the strain, so we added the growth curve of the most important engineered strains involved in three

parallel pathways, as shown in the revised Supplementary Fig. 2. In addition, we added the related description in the revised manuscript (Line 123). This growth curve could more directly show the impact of metabolic modification on engineered strains. In particular, the mutation of FPPS significantly reduced final OD₆₀₀ which might be due to the affect the biosynthesis significant biomass component ergosterol.

Reviewers' Comments:

Reviewer #3:

None